# Exploring the Value of Nodes with Multicommunity Membership for Classification with Graph Convolutional Neural Networks

Michael Hopwood [1,2,†] , Phuong Pho [1,†] and Alexander V. Mantzaris [1,*]

1   Department of Statistics and Data Science, University of Central Florida, Orlando, FL 32816, USA; michael.hopwood@knights.ucf.edu (M.H.); phuong.pho@ucf.edu (P.P.)
2   Sandia National Laboratories, Albuquerque, NM 87123, USA
*   Correspondence: alexander.mantzaris@ucf.edu; Tel.: +1-407-823-3631
†   Michael Hopwood and Phuong Pho contributed equally.

**Abstract:** Sampling is an important step in the machine learning process because it prioritizes samples that help the model best summarize the important concepts required for the task at hand. The process of determining the best sampling method has been rarely studied in the context of graph neural networks. In this paper, we evaluate multiple sampling methods (i.e., ascending and descending) that sample based off different definitions of centrality (i.e., Voterank, Pagerank, degree) to observe its relation with network topology. We find that no sampling method is superior across all network topologies. Additionally, we find situations where ascending sampling provides better classification scores, showing the strength of weak ties. Two strategies are then created to predict the best sampling method, one that observes the homogeneous connectivity of the nodes, and one that observes the network topology. In both methods, we are able to evaluate the best sampling direction consistently.

**Keywords:** graph convolutional neural networks; social networks; network science; citation networks; community labeling

## 1. Introduction

From the general study of networks [1] it can be seen how they are ubiquitous in many aspects of nature and even in the higher level networks of financial associations. Complex processes that have been studied [2] display behaviors that are related to their network structure (topology). One of the most famous mathematical investigations is one of the first encounters with a problem statement derived from a network (graph structure), the 'Seven Bridges of Königsberg problem' [3]. This question has spawned many intriguing questions, such as that of the traveling salesman problem [4].

In many situations, the total number of nodes for a network can be very large and treating each node as a unique ID does not allow for an investigation to reveal generalizations that can help in situations of inferring missing data, for instance. Simplifying these network nodes into a smaller set of labels can be done in various ways and is typically associated with community detection, where the connectivity of the nodes in the network directs the community memberships. This labeling process assists in the effort to simplify the node set. Each label applies some level of generalization with respect to aggregated behaviors across the allocated group labels. Examples for the label application can cover voting patterns that are pigeonholed into a small number of choices, consumer buying patterns in respect to certain products, and even different psychological profiles. This concept is applied in algorithms using collaborative filtering [5] (usually for retail), where recommendation systems apply a customer's interests to find the closest community set to predict an affinity for new items. The principle underlying the ability to group nodes together in this fashion relies upon a degree of homophily [6] in the groups. Examples of this are found in the work of [7], which studies how social network connections created

from friendships or interests can drive political engagements differently. For the growth of a network where edges are constructed, nodes create connections or affiliations and it becomes a question of determining the label for a node with which an edge is constructed. Choosing these connections becomes an important issue for the originating node as it produces label associations.

Methods such as logistic regression can infer the labels for nodes provided with the features after a training phase but does not model the connectivity between nodes, which provides extra information that can augment the feature information. This is important for accuracy when there is not enough feature information for accurate label prediction. Community detection algorithms such as the Louvain algorithm [8] take into account the placement of a node in the network topology, but they do not take into account the node features for allocating the labels. The methods of Graph Neural Networks (GNNs) [9] provide a framework that combines both feature information and network information in order to make inferences on the labels applied to nodes.

The methodology of the Simple Graph Convolutional Neural Network (SGC) [10] (described in more detail in the Methodology) presents an intuitive, simple, and expressive formulation for learning these latent representations of the node labels that builds upon the general theory of graph convolutional networks [11]. This methodology is appealing because the operations are linear between the adjacency matrix, the features, and the parameters prior to the use of the softmax function. This makes it an ideal candidate to work with in exploring different applications of its formulation as the feature projections are linear and the adjacency matrix is clearly an operation aggregating feature information of the vicinity of the nodes.

A machine learning pipeline usually consists of multiple parts: sample, explore, modify, model, and assess [12]. As the data travels through the pipeline, error propagates; a mistake in an earlier step may have resounding impacts on the pipeline's performance. Active learning focuses on the sampling step by prioritizing samples that are assumed to be of more aid to the task [13]. Sampling the training corpus prior to training the model is conducted to attempt to receive similar, if not better, accuracy while utilizing less data. This process has shown to be successful in multiple domains, like natural language processing and image data [14–17]. Recent work has been conducted on the performance of active learning on graph data [18,19]. In general, deep learning has shown great advantages in many fields [20–22] including spatial components for a complex object [23].

For graph data, there is a variety of node ranking algorithms such as pagerank [24] and voterank [25] that can be utilized to select nodes in the active learning process. In this paper, we look for optimal sampling methods for the node classification task across five real graph datasets that span a large domain of network topologies. Our main contributions is the discovery of the effect of the sampling method or direction (i.e., ascending versus descending selection) on the results of a node classification task.

This paper looks at how the SGC can be used to arrive at correct label allocations of nodes in networks (where nodes contain features) where the full set of network nodes is not provided. The context is that often the full network is not visible to the investigator and the set of nodes are actually sampled. From the investigation, it can be seen that how these nodes are sampled can change the accuracy of labels trying to be predicted. It could be assumed that nodes that have the largest number of edges would be the best representatives of a community label as their label would propagate to other nodes for whom it is the most central node. The investigation here counter-intuitively shows in the results that low ranked nodes in terms of centrality can provide better information on the labels. The fact that lower ranked central nodes contain more accurate information for the label allocation supports the general idea that 'weak-ties' are valuable [26]. Two strategies are proposed to predict the best sampling methods based on network topology. First, we find that the skewness of homogeneous connectivity distribution is an accurate predictor for sampling direction. Additionally, we empirically find a correlation between the topology structure, consolidated by a single statistic, and the sampling direction.

Section 2 delineates the methodology, including descriptions of the data, sampling methods, and the employed graph convolutional neural network. Section 3 shows the results along with a discussion of the results. Lastly, Section 4 describes the final takeaways and some potential future work.

## 2. Methodology

### 2.1. Data

An attributed graph $G = (X, A, y)$ is represented by three components: an adjacency matrix $A \in \mathbb{R}^{N \times N}$, a feature matrix $X \in \mathbb{R}^{N \times D}$, and a node label vector $y \in \mathbb{R}^N$. Real datasets were gathered from online resources; seven of the nine datasets were accessed using open-source python libraries [27,28]. The other two, Lastfm-Asia and Deezer-Europe, were downloaded from the Stanford Network Analysis Project's repository [29].

### 2.2. Sampling Methods

Two procedures of sampling are considered in this study, namely, descending and ascending. In the descending sampling, training instances are selected by gradually acquiring them from the most important nodes to the least important ones. On the contrary, ascending sampling gradually selects training samples starting from the least important nodes to the most important ones.

Three different criteria are used to evaluate a node's importance (centrality) for sampling orders.

#### 2.2.1. Degree

In degree sampling, we acquire nodes for training based on their corresponding number of directly connected neighbors (i.e., node's degree).

#### 2.2.2. PageRank

The PageRank algorithm [24] derives a web page's (node)'s rank by accumulating its incoming neighbors' ranks proportionally to their total number of outgoing connections. The resulting ranking represents the relative importance of pages in the network. In this study, we apply PageRank to rank all the nodes in our graphs and then sample them based on their rankings.

#### 2.2.3. VoteRank

The VoteRank algorithm [25] iteratively selects a set of important nodes called spreaders using voting scores given by the neighboring nodes. Once a node is selected as a spreader, it is excluded from the next round of voting and its direct neighbors' voting abilities are also reduced. In this study, we employ VoteRank to all nodes in the graph (by setting the number of spreaders as the total number of nodes) and then sample them based on their rankings.

### 2.3. Simple Graph Convolution (SGC)

SGC [30] is a simplified GNN model developed from GCN [31] by removing nonlinear activation functions between hidden layers and reparametrizing successive layers into one single layer. This simplification reduces the superfluous complexity of the GCN while retaining superb performance on many downstream tasks. The work of [32] illustrates SGC's expressive power on a node classification task and proposes a flexible regularization methodology to reduce the number of parameters and highlight a sparse set of important features. The SGC is a 'one-shot' learner, which simplifies the training procedure and allows for the full set of data points to be used for the parameter inference.

In this section, we briefly present the original SGC. An attributed graph dataset contains a graph $G = (V; \mathbf{A})$ and a feature matrix $X \in \mathbb{R}^{N \times D}$. The graph $G$ composed of $V = (v_1, v_2, \ldots, v_N)$ is a set of $N$ nodes (vertices); $\mathbf{A} \in R^{N \times N}$ is the adjacency matrix, where each element $a_{ij}$ represents an edge between node $v_i$ and $v_j$ ($a_{ij} = 0$ if $v_i$ and $v_j$ are

disconnected). We define the degree matrix $\mathbf{D} = \mathrm{diag}(d_1, d_2, \ldots, d_N)$ as a diagonal matrix whose off-diagonal elements are zero and each diagonal element $d_i$ captures the degree of node $v_i$ and $d_i = \sum_j a_{ij}$. Each row $x_i$ of the feature matrix $X \in \mathbb{R}^{N \times D}$ is the feature vector measured on each node of the graph. Each node $i$ receives a label from $C$ classes and hence can be coded as one hot vector $y_i \in \{0, 1\}^C$.

The GCNs and SGC add self-loops and normalize the adjacency matrix to get the matrix $\mathbf{S}$:

$$\mathbf{S} = \tilde{\mathbf{D}}^{-\frac{1}{2}} \tilde{\mathbf{A}} \tilde{\mathbf{D}}^{-\frac{1}{2}}, \tag{1}$$

where $\tilde{\mathbf{A}} = \mathbf{A} + \mathbf{I}$ and $\tilde{\mathbf{D}} = \mathrm{diag}(\tilde{\mathbf{A}})$. This normalization allows successive powers of the matrix to not influence the overall size of the projections. The SGC removes nonlinear transformation from the $k$-th-layer of the GCN resulting in a linear model of the form:

$$\hat{\mathbf{Y}} = \mathrm{softmax}(\mathbf{S} \ldots \mathbf{S}\mathbf{S}\mathbf{X}\mathbf{\Theta}^{(1)}\mathbf{\Theta}^{(2)} \ldots \mathbf{\Theta}^{(K)}). \tag{2}$$

The SGC classifier is then achieved by collapsing the repetitive multiplication of matrix $\mathbf{S}$ into the $k$-th power matrix $\mathbf{S}^K$ and reparametrizing the successive weight matrices as $\mathbf{\Theta} = \mathbf{\Theta}^{(1)}\mathbf{\Theta}^{(2)} \ldots \mathbf{\Theta}^{(K)}$; its structure as a GNN is defined by

$$\hat{\mathbf{Y}} = \mathrm{softmax}(\mathbf{S}^K \mathbf{X} \mathbf{\Theta}). \tag{3}$$

The parameter $k$ corresponds to the number of 'hops', which is the number of edge traversals in the network adjacency matrix $\mathbf{S}$. $k$ can be thought of as accumulating information from a certain number of hops away from a node (as described visually in [30]). If $k = 0$, the methodology becomes equivalent to a logistic regression application, which is known to be scalable to large datasets. Since the SGC introduces the matrix $\mathbf{S}$ as a linear operation, the same scalability applies. The weight matrix $\mathbf{\Theta}$ is trained by minimizing the cross entropy loss:

$$\mathcal{L} = \sum_{l \in \mathcal{Y}_L} \sum_{c \in C} Y_{lc} \ln \hat{Y}_{lc}, \tag{4}$$

where $\mathcal{Y}_L$ is a collection of labeled nodes. This model allows for a very computationally efficient exploration of the network-based datasets but this multilayer approximation may not provide the full extent of deep learning generalizations.

### 2.4. Evaluation of Network Topology

The network topology was evaluated using the coefficient of variation of the node's degree distribution.

$$CV_d = \frac{\mu_d}{\sigma_d}, \tag{5}$$

where $\mu_d = \frac{1}{N} \sum_{i=1}^{N} d_i$ is the average degree and $\sigma_d = \frac{1}{N-1} \sum_{i=1}^{N} (d_i - \mu)^2$ is the standard deviation of degree.

A low value of $CV_d$ occurs for networks that have high variation in their degree distributions compared to the mean degree. It indicates that important hubs (nodes) are highly connected to other nodes. On the contrary, a high value of $CV_d$ results from relatively low variation in degree distribution compared to the mean degree, where important nodes tends to be less popular.

The node degree centrality is defined by

$$D_i = \frac{d_i}{max(d_i)}, \tag{6}$$

where $d_i$ is the degree of node $i$. The homogeneous connectivity is the proportion of homogeneous connections that a node has normalized by its total number of connections.

$$\Omega_i = \frac{h_i}{d_i},\tag{7}$$

where $h_i$ is the number of homogeneous nodes within one degree.

### 3. Results

Sampling the nodes from a graph causes a change in performance. Active learning [13], as opposed to passive learning, selects a set of most-informative instances for training to achieve the best performance while using minimal samples. This paper utilizes common sampling methods (i.e., degree, PageRank, VoteRank) to select nodes in a graph for the training corpus using an ascending (i.e., lowest to highest score) and descending (highest to lowest score) fashion.

Two separate methods are formulated to predict the best sampling direction. A method that utilizes the network topology looks for a partition in the $CV_d$ domain. A positive to this method is it utilizes information that is readily available prior to classification, excluding the requirement of knowing the ground truths. A second method is proposed that compares the skewness of the homogeneous connectivity distributions to evaluate the best sampling direction. This method utilizes the ground truth to explain why the best sampling direction is, in fact, the best.

Two different types of plots are visualized for each dataset. The sampling result plot shows the performance of the node classification task (measured by accuracy) on various training sizes across multiple sampling techniques. The accuracy curves tends to improve as more training samples are recruited. The box-plots show results of random sampling with 10 replications. With further inspection, sampling methods (plotted as lines) can be viewed to pass specific judgment regarding its performance. If these sampling methods show better performance than random selection, it can be concluded that the method is an improvement.

The degree centrality plot is composed of a scatter plot illustrating the relationship between homogeneous connectivity, $\Omega_i$ (Equation (7)) node centrality (Equation (6)) (on the left panel), and a histogram showing distribution of homogeneous connectivity (on the right panel). The purple points describe the nodes that are sampled following the "degree ascending" method, when $s = 0.5$; likewise, the yellow points describe the nodes sampled following the "degree descending" method, when $s = 0.5$. As the scatterplot is visualized according to node degree centrality on the x-axis, a clear partition is found between the purple and yellow points. It is also informative to look at the histogram, which shows the distribution of homogeneous connectivity according to the sampling direction. On this plot, some deductions can be made as to why the superior sampling direction was more effective.

In the high $CV_d$ graph (i.e., Cora and Citeseer), all three descending methods almost uniformly render higher accuracies than the ascending methods across training sizes (Figure 1). The dominance of descending sampling in these graphs could be explained by the fact that important (central) papers of certain disciplines are usually cited by many papers in that same discipline. Consequently, the most important nodes contain crucial information about the class label and hence, are beneficial for the node classification task.

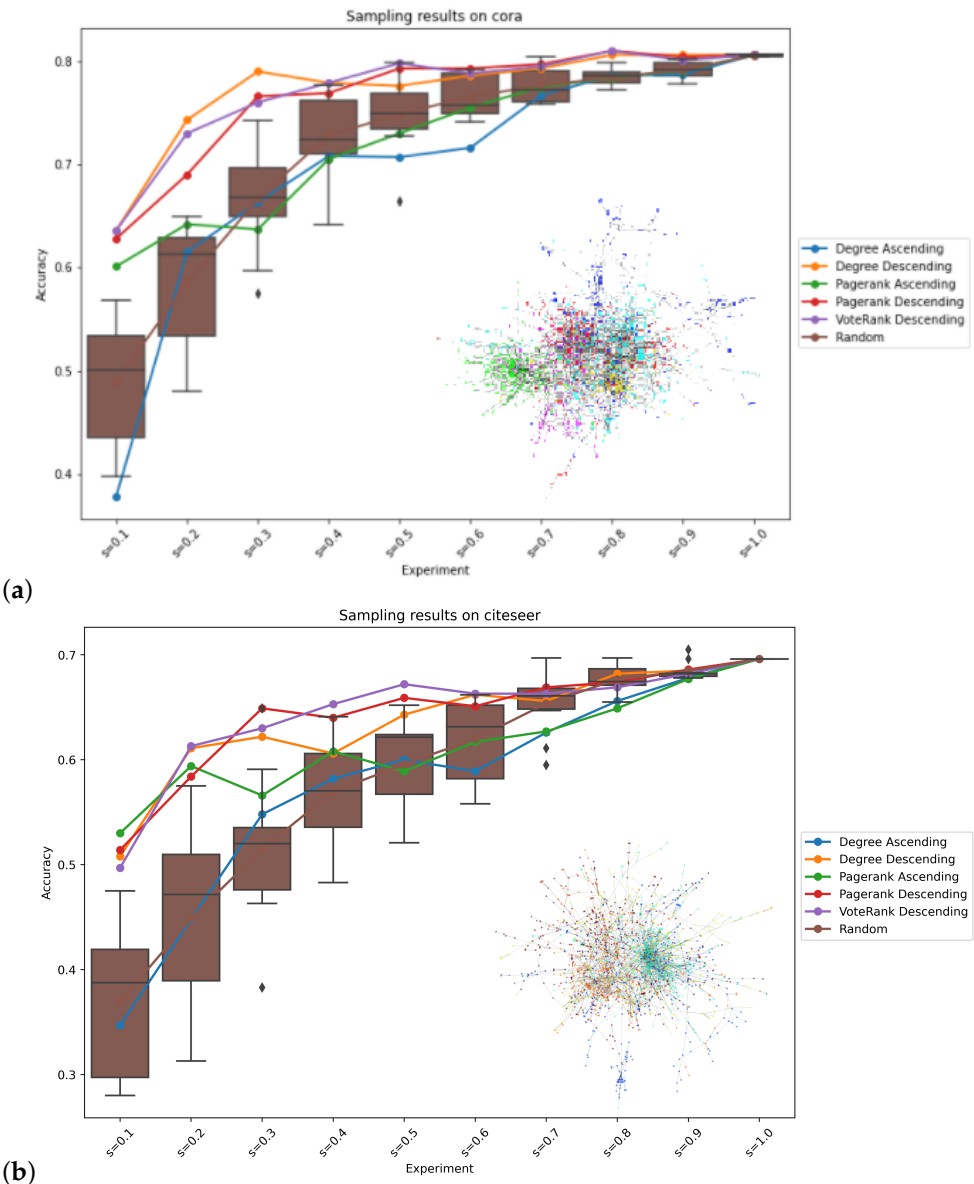

**Figure 1.** Here is presented the accuracy of the class predictions for the network node sample ratio using different node selection strategies. The large values of $CV_d$ for the network (Cora (**a**), Citeseer (**b**)) graph has higher accuracies when sampling nodes from the highest score to lowest score (i.e., 'descending' methods), showing the effectiveness of the node-ranking algorithms on a node classification task.

The distribution of homogeneous connectivity across the Cora and Citeseer data sets are similar (Figure 2). Homogeneous connections of both data sets exhibit left-skewed patterns, indicating the existence of clusters of informative nodes (high $\Omega$) and noisy nodes (low $\Omega$). Nodes with low $\Omega$ mainly connect with neighbors across different categories while nodes with high $\Omega$ mostly connect to neighboring nodes within the same category.

On Cora, the ascending and descending samplings possess a similar number of the most informative nodes. However, the bottom 50% of central nodes shows higher left skewness (Figure 2a). It indicates that less popular papers are noisier since they tend to get cited by papers in different categories. Hence, recruiting these samples in the training step is not desirable since they provide noisy representations of the corresponding categories and deteriorate the performance of the classifier.

On Citeseer, a different pattern occurs where large numbers of noisiest nodes exist in both sampling schemes. However, the descending samplings contain higher numbers of

moderate to high informative nodes as the distribution of the top central nodes exhibits a lower degree of left skewness (Figure 2b). Hence, descending sampling tends to work better since recruiting popular papers provides a smoother representation of their categories.

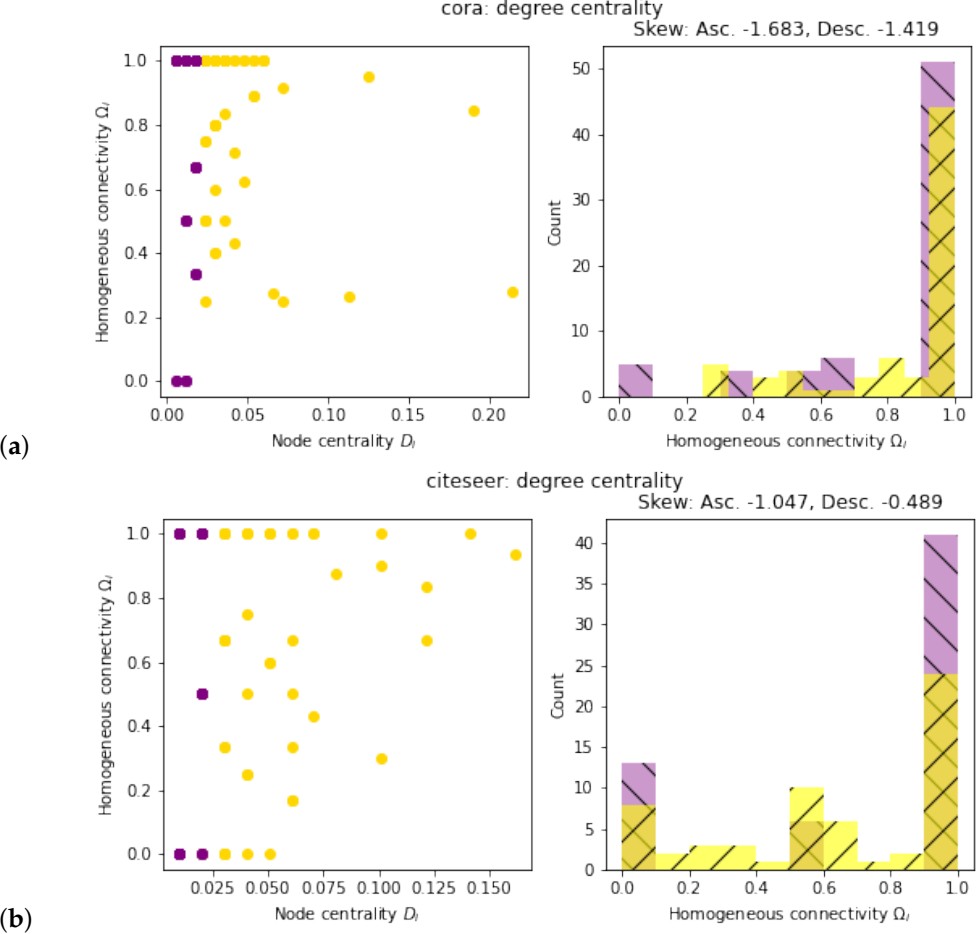

**Figure 2.** These plots present the investigation of datasets Cora (**a**) and Citeseer (**b**) to see the association of the centrality and the within or between edge class category edges contained. The left panels here present scatter plots of node degree centrality, $D_i$ against node homogeneous connectivity $\Omega_i$ on the training data. The upper half of nodes according to their centrality are colored in yellow while the lower half is presented in purple. The histogram on the right visualizes the distribution of homogeneous connections. The skew for each subset's distribution is annotated above the right graph.

Alternatively, we observe an opposite trend in low $CV_d$ graphs (i.e., Pubmed), where ascending samplings prevail Figure 3. The Pubmed citation graph contains publications about diabetes and hence, has a smaller scope compared with other citation data sets. Important (central) papers might get cited by other papers across classes due to the close nature of their categories. Therefore, important nodes contains a less differentiating factor for classification tasks. On the other hand, less important nodes might contain unique characteristics of the class and render useful information for the node classification task.

Pubmed's homogeneous connectivity distributions are highly left-skewed (Figure 4). Both sampling schemes contain a relatively high number of informative as well as noisy nodes. Descending sampling has relatively higher skewness implying a heterogeneous selection of high-quality and low-quality popular papers (in terms of their homogeneous connectivity). Popular papers (high $D$) with a low amount of within-category citations (low $\Omega$) get cited by other papers from different categories. Hence, the descending strategy has worse performance since these low-quality popular papers inevitably induce a confusing representation of the category.

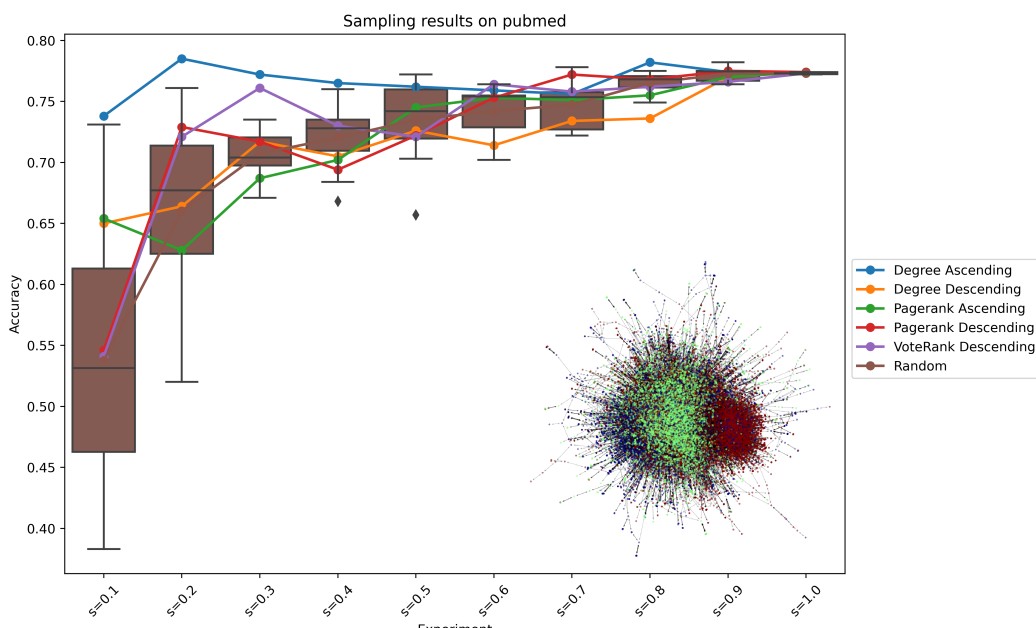

**Figure 3.** Low $CV_d$ networks (Pubmed) graphs have lower accuracies when sampling nodes from the lowest to highest score (i.e., 'ascending' methods), showing that the ranking algorithms are inversely beneficial to the node classification task.

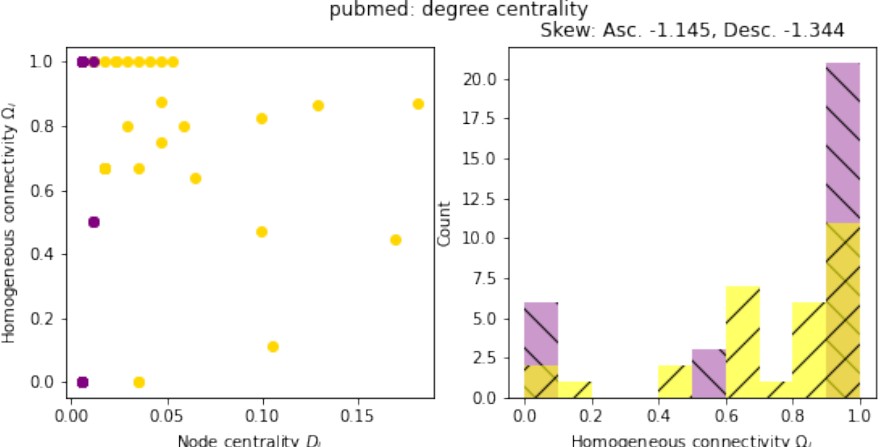

**Figure 4.** The left figure here presents scatter plots of node degree centrality $D_i$ against node homogeneous connectivity $\Omega_i$ on the training data. The upper half of nodes according to their centrality are colored in yellow while the lower half is presented in purple. The histogram on the right visualizes the distribution of homogeneous connections. The skew for each subset's distribution is annotated above the right graph.

The LastFM-Asia is a social media dataset that categorizes users based on their country of origin (Table 1). Node classification results change dramatically with training size, *s*. After $s = 0.3$, the ascending sampling methods consistently perform better than descending methods. In this social media, nodes with smaller importance are more indicative of a node's label—the person's country of origin. Users with smaller followerships are more likely to be connected with people they know personally, within their real-life circle. However, people with more followers are more famous, and likely have more followers across the globe, therefore causing the country to be hard to discern. Much like the other datasets, the homogeneous connectivity distribution for LastFM-Asia is left-skewed. In other words,

there exists a large set of edges that are interconnected within the community and fewer connected to other communities. The skewness of the ascending selections is greater than that of the descending selections, therefore, a conclusion is made towards the utilization of ascending as the sampling direction, which matches with the node classification results.

**Table 1.** Dataset information, like network structure and domain-specific definitions.

| Dataset | Ref. | #Nodes #Edges #Classes | Description |
|---|---|---|---|
| Cora | [33] | 2708 5278 7 | Scientific publications (nodes), defined by a binary vector indicating the presence of words in the paper (features), connected in a paper citation web (edges), and categorized by topic (labels). |
| Citeseer | [34] | 3327 4614 6 | Scientific publications (nodes), defined by a binary vector indicating the presence of words in the paper (features), connected in a paper citation web (edges), and categorized by topic (labels). |
| Pubmed | [35] | 19,717 44,325 3 | Diabetes-focused scientific publications (nodes), defined by a binary vector indicating the presence of words in the paper (features), connected in a paper citation web (edges), and categorized by topic (labels). |
| Amazon-PC | [36] | 13,752 287,209 10 | Computer goods sold at Amazon (nodes), defined by a bag-of-words encoded vector of the product's reviews, connected with groups of products that are frequently bought together (edges), and grouped into product categories. |
| Amazon-Photo | [36] | 7650 143,663 8 | Photos sold at Amazon (nodes), defined by a bag-of-words encoded vector of the product's reviews, connected with groups of products that are frequently bought together (edges), and grouped into product categories. |
| Coauthor-CS | [27] | 163,788 18,333 15 | Authors (nodes) of computer science papers, defined by a vector of keywords in their published papers, connected by coauthorship (edges), and categorized by the author's most active field of study. |
| Coauthor-Physics | [27] | 34,493 495,924 5 | Authors (nodes) of physics papers, defined by a vector of keywords in their published papers, connected by coauthorship (edges), and categorized by the author's most active field of study. |
| Lastfm-Asia | [33] | 7624 27,806 18 | Social network users (nodes) using LastFM, defined by their artists-of-interest, connected by their mutual followers (edges), and categorized by the user's location. |
| Deezer-Europe | [33] | 28,281 92,752 2 | Social media users (nodes) using Deezer, defined by their artists-of-interest, connected by mutual followers (edges), and categorized by gender. |

The Deezer-Europe social media dataset shows varied results, Figure 5. Most of the sampling methods, agnostic to the sampling direction, consistently perform better than random. In other words, the more-popular and less-popular nodes are helpful in the node gender-classification task, as opposed to users in the middle-ground. Intuitively, less popular users' networks are likely respond to gender homophily, as shown in certain age groups in [37]. More popular users likely have growing followerships that can be based on mutual interests, especially in this network's musical context. The Deezer-Europe dataset renders a unique homogeneous connectivity distribution, showing one that appears Gaussian. The average value is centered near 0.50. The skew of the $\Omega_i$ domain is in favor of the descending sampling process, which is also the conclusion made by the $CV_d$ process. This is shown in Figure 6 for the same datasets.

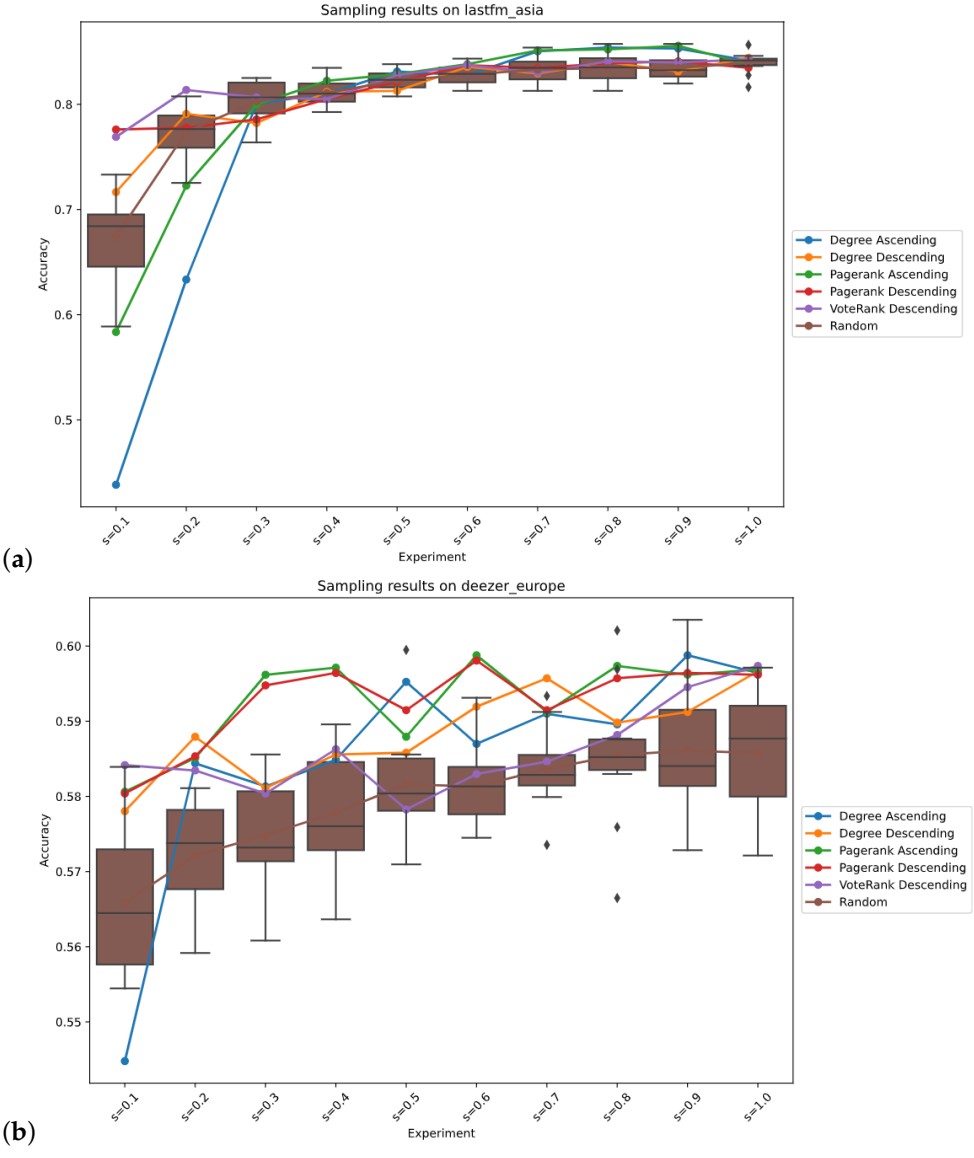

**Figure 5.** Here the accuracy of the model for the node percentages samples is presented using the different node directions on LastFm data (**a**) and Deezer-Europe data (**b**). The performance of this pipeline on the Deezer_Europe social media dataset (plot **b**) is unusual in that almost all sampling methods are uniformly better than random selection.

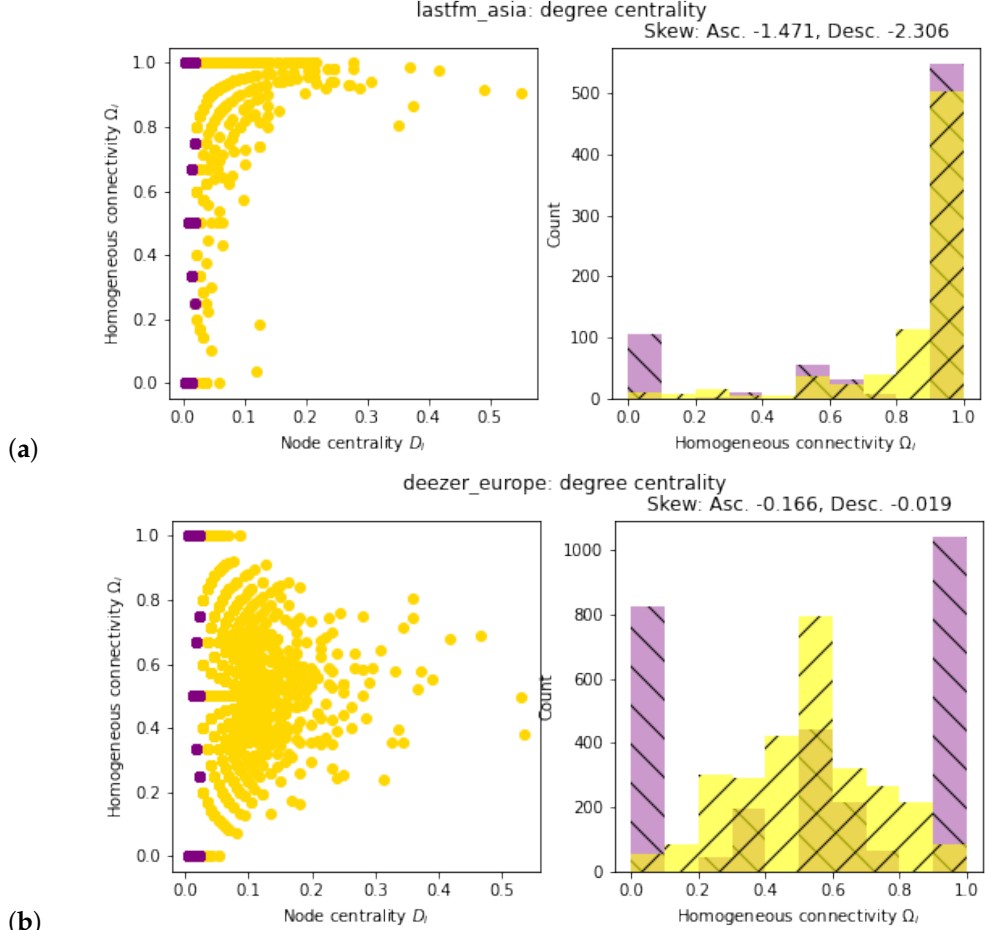

**Figure 6.** The left figure here presents scatter plots of node degree centrality $D_i$ against node homogeneous connectivity $\Omega_i$ on the training data. The upper half of nodes according to their centrality are colored in yellow while the lower half is presented in purple. The histogram on the right visualizes the distribution of homogeneous connections. The skew for each subset's distribution is annotated above the right graph. (**a**) LastFm; (**b**) Deezer-Europe.

In this paper, we have found a correlation between network topology and the optimal sampling strategy. Interested readers are referred to the AppendixA section for results of the remaining datasets that are not discussed here. This fact implies that machine learning practitioners can deduce an optimal sampling strategy by (1) evaluating their network topology and (2) observing its position on Figure 7. The results show that no sampling method is superior in terms of accuracy; the logistic probability of a descending sampling evaluation producing the best results increases with an increasing $CV_d$. A high $CV_d$ occurs for highly connected graphs (high $\mu_d$) where all nodes have a similar number of connections (low $\mu_d$). Coauthor-cs, citeseer, and deezer_europe are among the highest scorers in $CV_d$. In these graphs, important nodes have relatively lower popularity, which correlates with a descending sampling direction because these nodes contain defining characteristics for their associated categories. A low $CV_d$ occurs for low connected graphs where important nodes are highly connected, i.e., more popular, which correlates with an ascending direction. Sampling less popular nodes are more beneficial since they contain distinct characteristics to represent associated categories.

Network-topology-informed sampling methods (i.e., all methods except random) seem to perform well on the node classification tasks, often resulting in similar accuracies while utilizing a smaller amount of data. Additionally, independent of the ascending/descending, we see across the board a higher number of cases where the more complicated sampling procedures (i.e., Pagerank/Voterank) outperform Degree. While

we see an increase in performance, there is a trade-off with computation time; nodes' degree distribution can be computed swiftly while Pagerank and VoteRank require complex evaluation and hence, are more computationally expensive.

The skewness of the homogeneous connectivity $\Omega_i$ distribution is indicative of the better performing sampling direction. Left-skew distribution is more common because it is expected that there is an association between network topology and nodes' labels, in which nodes with the same label tend to connect with each other. Graph neural networks utilize message passing to learn expressive node embedding for a given task [38]. The mechanism involves aggregating features of a node's neighbors to produce a smoother representation, where neighboring nodes tend to have a similar property such as belonging to the same class. Therefore, a left skew of $\Omega_i$ is suitable for graph neural networks to learn the effective node embedding for a classification task.

Under the assumption that a $\Omega_i$ is left-skewed, the sampling method that renders a weaker skewness will be the one that performs better. A stronger left-skewed distribution has an elongated tail, which recruits more noisy, low-information nodes. These samples aggregate features of neighboring nodes belonging to other classes and provide a poor representation of their own classes. Their noisy representations inevitably induce more confusion to the model and degrade performance on the classification task. Table 2 demonstrates the agreement between skewness of the homogeneous distribution and the best sampling approaches.

**Table 2.** The sampling direction is predicted with a high accuracy by studying the skewness of the homogeneity connectivity distribution. The misclassifications are likely caused by a lack of node importance evaluators, which are robust to graph topology.

| Dataset | Prediction | Actual |
| --- | --- | --- |
| Cora | Descending | Descending |
| Citeseer | Descending | Descending |
| Pubmed | Ascending | Ascending |
| Amazon-pc | Descending | Ascending |
| Amazon-photo | Ascending | Ascending |
| Coauthor-cs | Descending | Descending |
| Coauthor-physics | Ascending | Descending |
| Lastfm_Asia | Ascending | Ascending |
| Deezer_Europe | Descending | Descending |

Some graphs (i.e., amazon-pc, coauthor-physics) do not robustly fit the node importance evaluators utilized in this study, as indicated by the poor performance of informed samplers compared to random sampling. Both examples show conflicting results when utilizing our two sampling direction detection schemas; coauthor-physics concludes ascending via $\Omega_i$ and descending via $CV_d$ and amazon-pc concludes descending via $\Omega_i$ and ascending via $CV_d$. Future work will be required to observe the domain at which this phenomena occurs as both examples of conflicting indications occur at the edges of the $CV_d$ domain. Given that node degree is one of the measures of node centrality, we would assume that using other centrality measurement (such as VoteRank) might render harmonious conclusions of sampling schemes from $\Omega_i$ and $CV_d$.

In practice, obtaining homogeneous connectivity distribution prior to sampling is impractical since it requires knowledge about the labels in the calculation process. Hence, we developed an alternative criteria to help practitioners select the best sampling approach based on the coefficient of variation of the node degree. Our experiments show a relationship between the network topology (summarized by $CV_d$) and the best sampling direction (Figure 7).

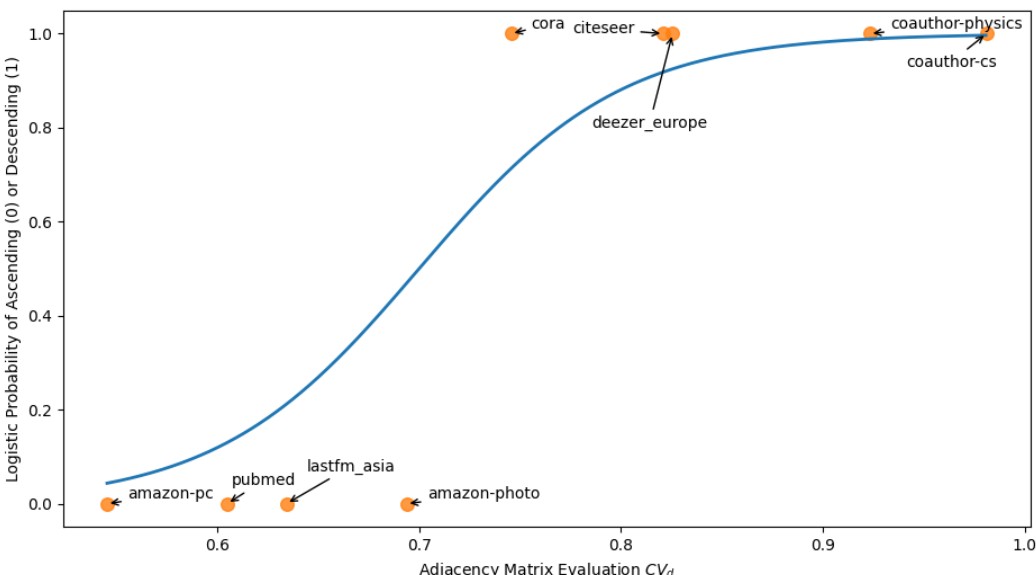

**Figure 7.** Logistic probability (blue line) shows an increasing likelihood of a descending sampling procedure as the coefficient of variance of the degree ($CV_d$) increases. Results show that a complete separation is defined by $CV_d$.

## 4. Discussion

The careful selection of nodes for a machine learning process helps increase the accuracy of correct label prediction, enticing the study of different sampling methods. Two sampling methods are evaluated, ascending and descending, in which samples are selected based on node centrality as defined by three different measures (Voterank, Pagerank, and degree). We conclude that there does not exist a uniformly best method for node selection across all network topologies.

Prior to building a classification pipeline, it is useful for the practitioner to have an estimate on which sampling direction is superior. An indicative measure of best sampling strategy is the skewness of homogeneous connectivity distribution. A left-skewed distribution is desirable since neighboring nodes tend to belong to the same class and, hence, produce smoother representation for the node classification task. However, we found that a strong left skewness—indicating a selection of more noisy and low informative nodes—is detrimental to the performance of the classification task. However, rendering the homogeneous connectivity is impractical for practitioners due to its reliance on knowing the node's labels. Therefore, we present a second method that only requires network topology information ($CV_d$). This method is empirically proven.

Future work will apply these findings to large social media networks for tasks like job searching. Further, applications to knowledge embedding in the natural language processing domain will be pursued.

**Author Contributions:** Conceptualization, M.H., P.P. and A.V.M.; methodology, M.H., P.P. and A.V.M.; software, M.H. and P.P.; validation, M.H., P.P. and A.V.M.; investigation, M.H., P.P. and A.V.M.; writing, M.H., P.P. and A.V.M.; visualization, M.H., P.P. and A.V.M.; supervision, A.V.M. All authors have read and agreed to the published version of the manuscript.

**Funding:** This material is supported by the U.S. Department of Energy's Office of Energy Efficiency and Renewable Energy—Solar Energy Technologies Office (under Agreement Number 34172 and as part of the Durable Modules Consortium (DuraMAT), an Energy Materials Network Consortium). Sandia National Laboratories is a multimission laboratory managed and operated by National Technology and Engineering Solutions of Sandia, LLC, a wholly owned subsidiary of Honeywell International, Inc., for the U.S. Department of Energy's National Nuclear Security Administration

under contract DE-NA-0003525. The views expressed in the article do not necessarily represent the views of the U.S. Department of Energy or the United States Government.

**Institutional Review Board Statement:** Not applicable.

**Informed Consent Statement:** Not applicable.

**Data Availability Statement:** The datasets used come from available sources referred to in the citations.

**Conflicts of Interest:** The authors declare no conflict of interest.

## Appendix A

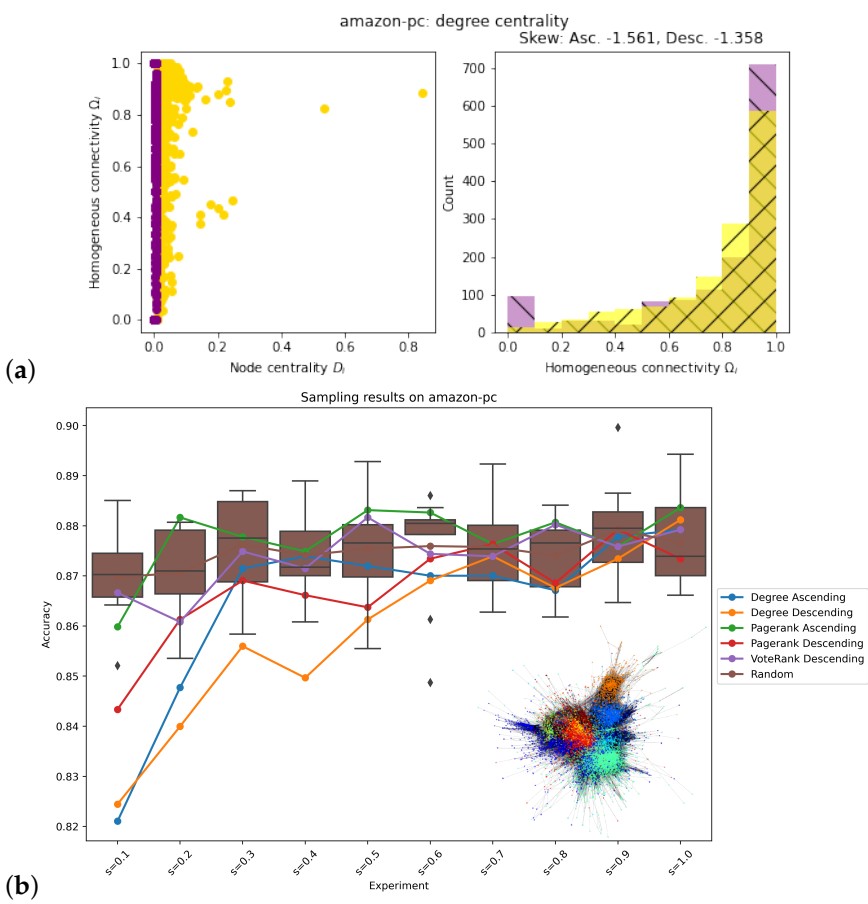

**(a)**

**(b)**

**Figure A1.** Panel (**a**) shows degree centrality plots where the left figure presents scatter plots of node degree centrality $D_i$ against node homogeneous connectivity $\Omega_i$ on the training data of amazon-pc. The upper half of nodes according to their centrality are colored in yellow while the lower half is presented in purple. The histogram on the right visualizes the distribution of homogeneous connections. The skew for each subset's distribution is annotated above the right graph. Panel (**b**) presents the sampling result on the training data of amazon-pc. Observing panel (**b**), random sampling appears almost uniformly superior.

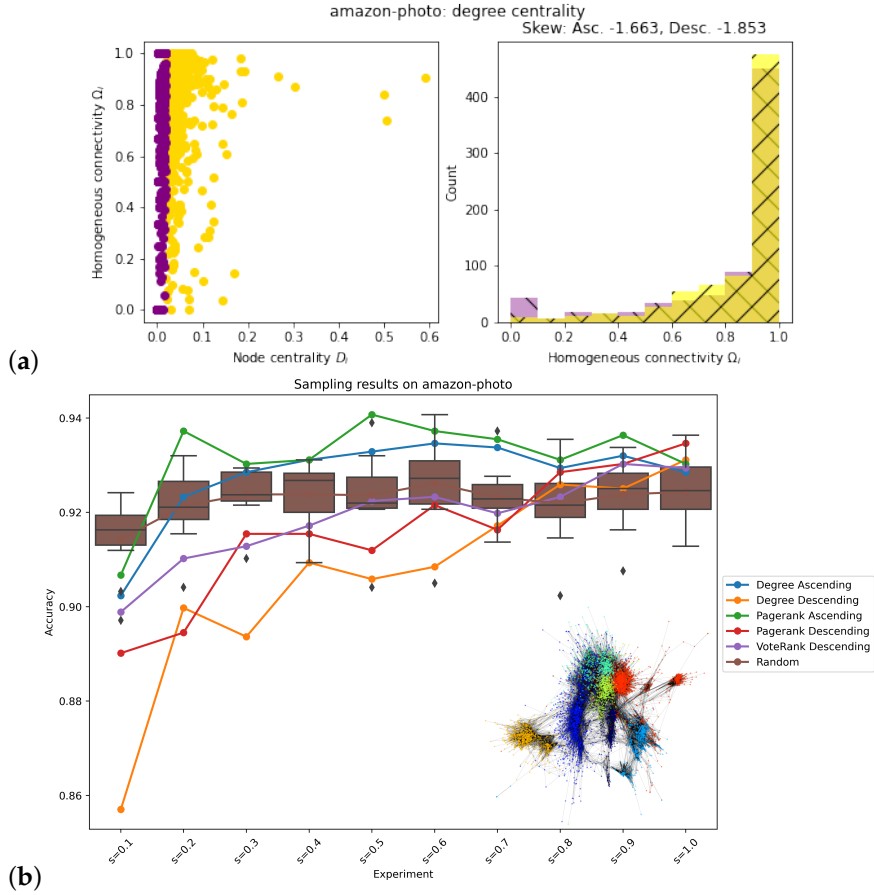

(**a**)

(**b**)

**Figure A2.** Panel (**a**) shows degree centrality plots where the left figure presents scatter plots of node degree centrality $D_i$ against node homogeneous connectivity $\Omega_i$ on the training data of amazon-photo. The upper half of nodes according to their centrality are colored in yellow while the lower half is presented in purple. The histogram on the right visualizes the distribution of homogeneous connections. The skew for each subset's distribution is annotated above the right graph. Panel (**b**) presents the sampling result on the training data of amazon-photo.

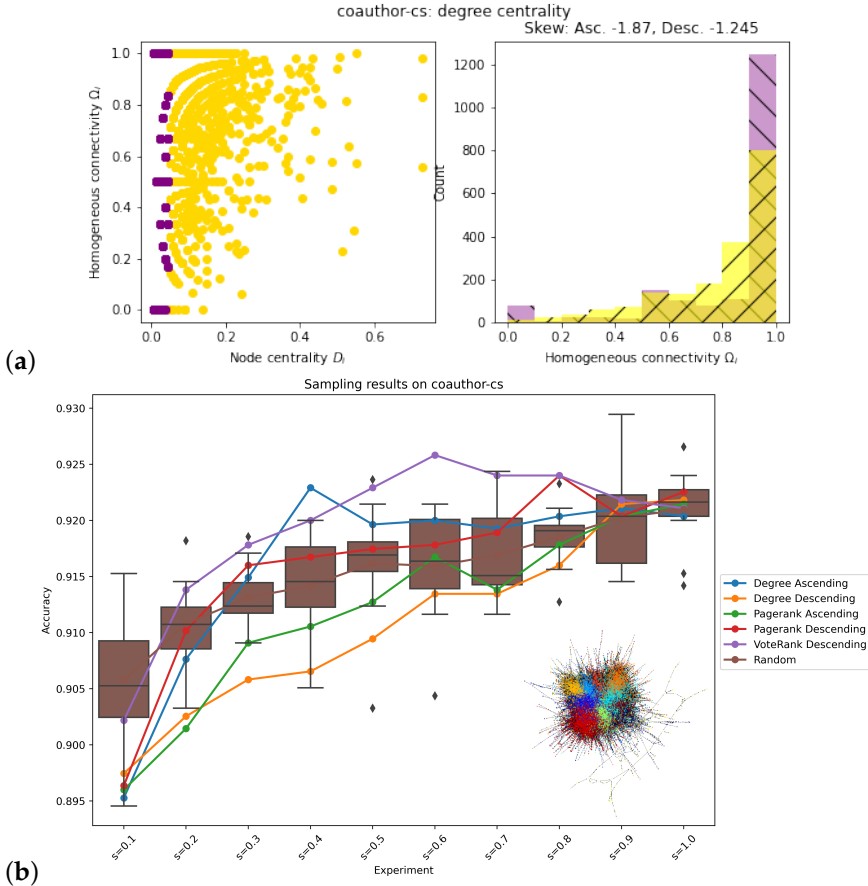

**(a)**

**(b)**

**Figure A3.** Panel (**a**) shows degree centrality plots where the left figure presents scatter plots of node degree centrality $D_i$ against node homogeneous connectivity $\Omega_i$ on the training data of coauthor-cs. The upper half of nodes according to their centrality are colored in yellow while the lower half is presented in purple. The histogram on the right visualizes the distribution of homogeneous connections. The skew for each subset's distribution is annotated above the right graph. Panel (**b**) presents the sampling result on the training data of coauthor-cs.

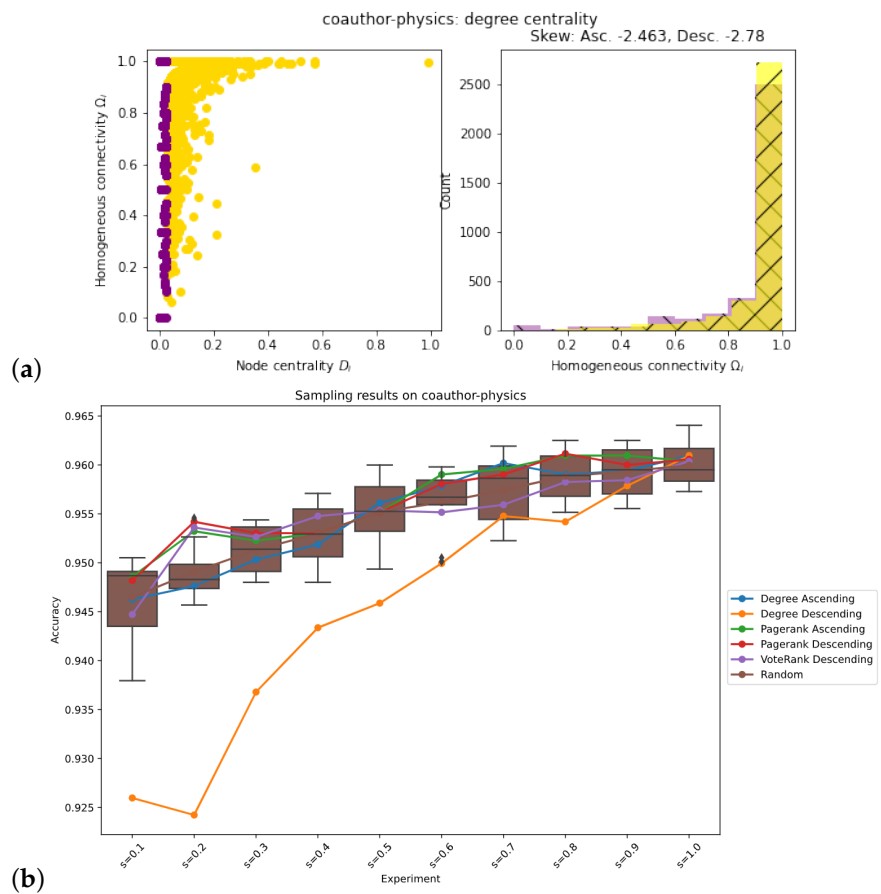

**(a)**

**(b)**

**Figure A4.** Panel (**a**) shows degree centrality plots where the left figure presents scatter plots of node degree centrality $D_i$ against node homogeneous connectivity $\Omega_i$ on the training data of coauthor-physics. The upper half of nodes according to their centrality are colored in yellow while the lower half is presented in purple. The histogram on the right visualizes the distribution of homogeneous connections. The skew for each subset's distribution is annotated above the right graph. Panel (**b**) presents the sampling result on the training data of coauthor-physics.

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
