# Peer review of "Exploring the Value of Nodes with Multicommunity Membership for Classification with Graph Convolutional Neural Networks"

_information, doi:10.3390/info12040170_

Round 1

Reviewer 1 Report

This paper still needs improvement before acceptance for publication. My detailed comments and suggestions are given as follows:

  1. Nowadays, deep learning has shown great advantages in many fields [1]. The authors should discuss the applications [2], challenges [3], and corresponding techniques [4] of deep learning, especially the generalization guarantee of deep learning models. The following papers on the research of generalization capability in the latest five years should be analyzed in the related work.

[1] “Improvement of generalization ability of deep CNN via implicit regularization in two-stage training process,” IEEE Access, 2018.

[2] “Deep Learning Strong Parts for Pedestrian Detection,” IEEE International Conference on Computer Vision, 2016.

[3] “Large scale deep learning for computer aided detection of mammographic lesions,” Medical Image Analysis, 2017.

[4] “Spectrum interference-based two-level data augmentation method in deep learning for automatic modulation classification,” Neural Processing & Applications, 2020. DOI: 10.1007/s00521-020-05514-1

  1. The specific structure of GNN should be presented.

  1. More comparative experiments should be added to illustrate the superiority of the propose method, and the experimental results should be further analyzed.

  1. Training details of GNN should be given in detail.

  1. Discussions about the generalization performance of deep learning model are encouraged.

  1. Why is it just used GNN instead of other models, such as random forest or CNN?

  1. More information about the experimental data set should be introduced.

Author Response

-These references which the reviewer has directed the authors have now been included.

 @article{zheng2018improvement,
  title={Improvement of generalization ability of deep CNN via implicit regularization in two-stage training process},
  author={Zheng, Qinghe and Yang, Mingqiang and Yang, Jiajie and Zhang, Qingrui and Zhang, Xinxin},
  journal={IEEE Access},
  volume={6},
  pages={15844--15869},
  year={2018},
  publisher={IEEE}
}

@inproceedings{tian2015deep,
  title={Deep learning strong parts for pedestrian detection},
  author={Tian, Yonglong and Luo, Ping and Wang, Xiaogang and Tang, Xiaoou},
  booktitle={Proceedings of the IEEE international conference on computer vision},
  pages={1904--1912},
  year={2015}
}

@article{kooi2017large,
  title={Large scale deep learning for computer aided detection of mammographic lesions},
  author={Kooi, Thijs and Litjens, Geert and Van Ginneken, Bram and Gubern-M{\'e}rida, Albert and S{\'a}nchez, Clara I and Mann, Ritse and den Heeten, Ard and Karssemeijer, Nico},
  journal={Medical image analysis},
  volume={35},
  pages={303--312},
  year={2017},
  publisher={Elsevier}
}

@article{zheng2020spectrum,
  title={Spectrum interference-based two-level data augmentation method in deep learning for automatic modulation classification},
  author={Zheng, Qinghe and Zhao, Penghui and Li, Yang and Wang, Hongjun and Yang, Yang},
  journal={Neural Computing and Applications},
  pages={1--23},
  year={2020},
  publisher={Springer}
}

A brief motivation for these citations is also provided. 

-"The specific structure of GNN should be presented."

 Included is this: ", and its structure as a GNN is defined by:" before the definition of the model in Eqn 3.

-"More comparative experiments should be added to illustrate the superiority of the propose method, and the experimental results should be further analyzed."

The authors agree with the benefit of including more data and experiments to illustrate the methodological exploration. 4 new datasets have now been used which all agree with the previous results. Those datasets are;

'amazon-pc' 'amazon-photo' from [Leskovec, J.; Krevl, A. SNAP Datasets: Stanford Large Network Dataset Collection.  http://snap.stanford.edu/data, 2014.]

'co-author cs' and 'co-author physics' from [Tian, Y.; Luo, P.; Wang, X.; Tang, X. Deep learning strong parts for pedestrian detection.  Proceedings of the IEEE internationalconference on computer vision, 2015, pp. 1904–1912]

"Training details of GNN should be given in detail."

Included this: "The SGC is a 'one-shot' learner which simplifies the training procedure and allows for the full set of data points to be used for the parameter inference." The nature of the SGC does greatly simplify the training procedure in comparison to approaches taken by methodological applications such as the ones the reviewer brought to our attention. 

-"Discussions about the generalization performance of deep learning model are encouraged."

The generalization of the model here is now touched upon at the end of the methodology.

-"Why is it just used GNN instead of other models, such as random forest or CNN?"

The value of the GNN is that it allows for an explicit use of the network adjacency information. If the CNN or a Random Forest had been used it would also have to learn the connectivity information as well. Here the GNN, using the 'A' variable which is the adjacency matrix does not need to learn it but knows the relational influence for the feature projections.  

-"More information about the experimental data set should be introduced."

More information is now provided in the table and the text body.

Reviewer 2 Report

The authors raise a very important subject. Especially in relation to such key issues related to e.g. advertising profiling and behavior analysis based on data and behavior left in social networks. The results of the authors' research can also be applied here.
There are many works and working methods in this field. However, it is a branch of knowledge that is constantly undergoing development due to the continuous modification of the implementation of social networks, but also the modification of users' behaviors resulting, inter alia, from their changing digital awareness.
The authors present the state of art in depth, which proves a good analysis of the state of art and shows that their current existence is not a coincidence.
The authors present their methodology in a transparent manner and adequate to the activities undertaken.
There are even some comments on the results obtained in the work. The obtained results can be useful in practice.
Interesting work but.
There is no appropriate publication format, i.e. IMRAD. Besides, the conclusions presented at the end are trivial. First of all, careful selection of nodes helps to increase the accuracy of the correct etiquette prediction. The conclusion that experience is useful for a better sampling direction estimate is also trivial. These conclusions could have been known and, in fact, are known without the research conducted by the authors.

Author Response

-"There is no appropriate publication format, i.e. IMRAD."

Previously the structure of the paper sections were; Introduction / Methodology / Results / Conclusions. The authors acknowledge the importance of following the consensus format recommendation and have made the changes to follow IMRAD. 

-"Besides, the conclusions presented at the end are trivial. First of all, careful selection of nodes helps to increase the accuracy of the correct etiquette prediction. The conclusion that experience is useful for a better sampling direction estimate is also trivial."

The reviewer is correct in stating that a 'careful selection of nodes helps to increase the accuracy of the correct etiquette prediction' is not a novel idea or concept being presented. It follows that it can also be expected that 'experience is useful for a better sampling direction estimate'. There are 2 key points which the paper does offer to the reader.

1. The exploration over the range of datasets, and the visual summaries of the results did not come without a considerable amount of effort. The authors wish to assist a potential reader by sharing these results for them to guide their own explorations without them having to repeat the same operations.

2. There are many alternative paths to explore this question that could have taken which do not produce the same quality of results. It also provides an addition to the body of literature exploring the application of the SGC which is a relatively new GNN model. How the SGC works in these situations has not, to our knowledge, been demonstrated. 

- "These conclusions could have been known and, in fact, are known without the research conducted by the authors."

The authors have looked for a reference to find an overlap where these conclusions are presented and not found it. The search for prior work that may use a different dataset or another Graph Convolutional Neural Network in a similar setting was made. Before beginning this work the body of literature was searched to find an approach that can help assist the direction of the sampling procedure, based upon network features; and it was not found.

Given this critism the authors recognize the deficiency in their research exposition since the merits of the methodological development have not been appreciated. Various sections have received modifications to highlight the key components of the description in the methodology and especially the results. 4 new datasets are now included and presented in Figure 7 which all coincide with the general pattern that governs the sampling approach to be taken. The use of the eqns 5-7 delivers a means to assess the network from a network science perspective for the use in an ML pipeline. The use of the skew for directing the sampling procedure and finding this to hold in a selection of datasets is novel from what the authors are aware of.

Most importantly it could be expected that the most central nodes of a community (topologically) would provide the most valuable information in terms of class identification for other nodes. This is not always the case, and the result that lower ranked nodes can be more useful is surprising. It is even more interesting that the skew of the node cross community link homogeneity can direct this order provides a novel conclusion. 

Round 2

Reviewer 1 Report

This paper can be accepted.